# Complexity-Guided Curriculum Learning for Text Graphs

**Nidhi Vakil**
Department of Computer Science
University of Massachusetts Lowell
nvakil@cs.uml.edu

**Hadi Amiri**
Department of Computer Science
University of Massachusetts Lowell
hadi@cs.uml.edu

## Abstract

Curriculum learning provides a systematic approach to training. It refines training progressively, tailors training to task requirements, and improves generalization through exposure to diverse examples. We present a curriculum learning approach that builds on existing knowledge about text and graph complexity formalisms for training with text graph data. The core part of our approach is a novel data scheduler, which employs "spaced repetition" and complexity formalisms to guide the training process. We demonstrate the effectiveness of the proposed approach on several text graph tasks and graph neural network architectures. The proposed model gains more and uses less data; consistently prefers text over graph complexity indices throughout training, while the best curricula derived from text and graph complexity indices are equally effective; and it learns transferable curricula across GNN models and datasets. In addition, we find that both node-level (local) and graph-level (global) graph complexity indices, as well as shallow and traditional text complexity indices play a crucial role in effective curriculum learning.

## 1 Introduction

Message passing (Gilmer et al., 2017) is a widely used framework for developing graph neural networks (GNNs), where node representations are iteratively updated by aggregating the representations of neighbors (a subgraph) and applying neural network layers to perform non-linear transformation of the aggregated representations. We hypothesize that topological complexity of subgraphs or linguistic complexity of text data can affect the efficacy of message passing techniques in text graph data, and propose to employ such complexity formalisms in a novel curriculum learning framework for effective training of GNNs. Examples of graph and text complexity formalisms are node centrality and connectivity (Kriege et al., 2020); and word rarity and type token ratio (Lee et al., 2021a) respectively.

In Curriculum learning (CL) (Bengio et al., 2009) data samples are scheduled in a meaningful difficulty order, typically from *easy* to *hard*, for iterative training. CL approaches have been successful in various areas (Graves et al., 2017; Jiang et al., 2018; Castells et al., 2020), including NLP (Settles and Meeder, 2016; Amiri et al., 2017; Zhang et al., 2019; Lalor and Yu, 2020; Xu et al., 2020; Kreutzer et al., 2021; Agrawal and Carpuat, 2022; Maharana and Bansal, 2022). Existing approaches use data properties such as sentence length, word rarity or syntactic features (Platanios et al., 2019; Liu et al., 2021); and model properties such as training loss and its variations (Graves et al., 2017; Zhou et al., 2020) to order data samples for training. However, other types of complexity formalisms such as those developed for graph data are largely underexplored. Recently, Wang et al. (2021) proposed to estimate graph difficulty based on intra- and inter-class distributions of embeddings, realized through neural density estimators. Wei et al. (2023) employed a selective training strategy that targets nodes with diverse label distributions among their neighbors as difficult to learn nodes. Vakil and Amiri (2022) used loss trajectories to estimate the emerging difficulty of subgraphs and weighted sample losses for data scheduling. We encourage readers to see (Li et al., 2023; Yang et al., 2023) for recent surveys on graph CL approaches.

We propose that existing knowledge about text and graph complexity can inform better curriculum development for text graph data. For example, a training node pair that shares many *common neighbors* is expected to be easier for link prediction than a local bridge[1] that lacks common neighbors. Motivated by this rationale, we present a complexity-guided CL approach for text graphs (TGCL), which employs multiview complexity formalisms to space training samples over time for iterative training. It advances existing research as follows:

---

[1] An edge that is not part of a triangle in the graph.

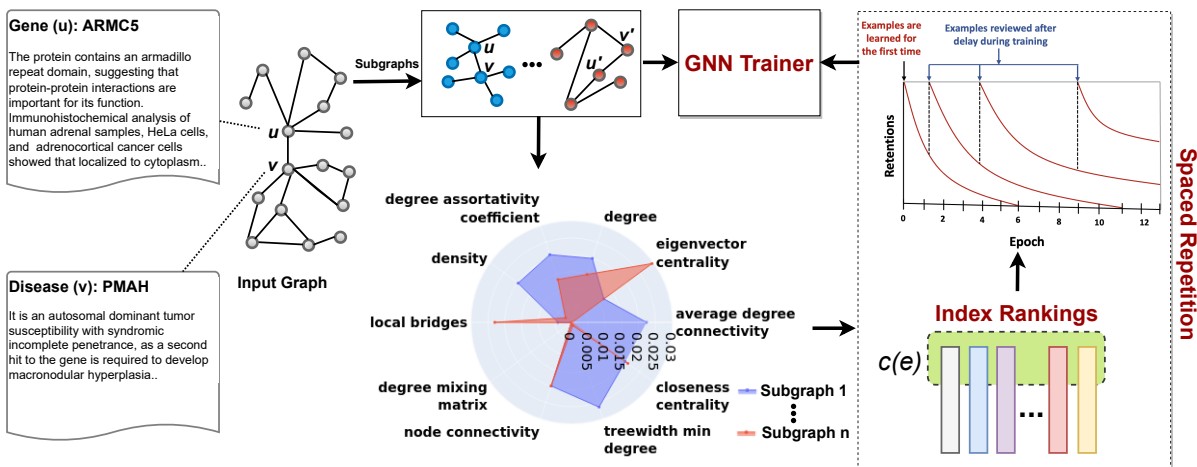

Figure 1: The architecture of the proposed model, TGCL. It takes subgraphs and text(s) of their target node(s) as input. The radar chart shows graph complexity indices which quantify the difficulty of each subgraphs from different perspectives (text complexity indices are not shown for simplicity). Subgraphs are ranked according to each complexity index and these rankings are provided to TGCL scheduler to space samples over time for training.

- a new curriculum learning framework that employs graph and text complexity formalisms for training GNNs on text graph data, and
- insights into the learning dynamics of GNNs, i.e., which complexity formalisms are learned by GNNs during training.

We conduct extensive experiments on real-world datasets and across GNN models, focusing on link prediction and node classification tasks in text graphs. The proposed model gains 5.1 absolute points improvement in average score over the state-of-the-art model, across datasets and GNN models, while using 39.2% less data for node classification than high-performing baselines. The results show that both node-level (local) and graph-level (global) complexity indices play a crucial role in training. More interestingly, although the best curricula derived from text and graph complexity indices are equally effective, the model consistently prefers text over graph complexity indices throughout all stages of training. Finally, the curricula learned by the model are transferable across GNN models and datasets[2].

## 2 Method

A curriculum learning approach should estimate the complexity of input data, determine the pace of introducing samples based on difficulty, and schedule data samples for training. As Figure 1 shows, TGCL tackles these tasks by quantifying sample difficulty through complexity formalisms (§2.1), gradually introducing training samples to GNNs

based on a flexible "competence" function (§2.2), and employing different data schedulers that order training samples for learning with respect to model behavior (§2.3). By integrating these components, TGCL establishes curricula that are both data-driven and model-dependent. In what follows, we present approaches for addressing these tasks.

### 2.1 Complexity Formalisms

**Graph complexity** (Kashima et al., 2003; Vishwanathan et al., 2010; Kriege et al., 2020) indices are derived from informative metrics from graph theory, such as node degree, centrality, neighborhood and graph connectivity, motif and graphlet features, and other structural features from random walk kernels or shortest-path kernels (Borgwardt and Kriegel, 2005). We use 26 graph complexity indices to compute the complexity score of data instances in graph datasets, see Table 1, and details in Appendix A.1. Since data instances for GNNs are subgraphs, we compute complexity indices for each input subgraph. For tasks involving more than one subgraph (e.g., link prediction), we aggregate complexity scores of the subgraphs through an order-invariant operation such as sum().

**Linguistic Complexity** Lee et al. (2021a) implemented various linguistics complexity features for readability assessment. We use 14 traditional and shallow linguistics indices such as Smog index, Coleman Liau Readability Score, and sentence length-related indices as text complexity indices in our study. See Appendix A.2 for details. We normalize complexity scores for each text and graph index using the L2 norm.

---

[2]Code and data are available at https://clu.cs.uml.edu/tools.html

| Degree based | Computing based |
|---|---|
| degree ⋆ | ramsey R2 ⋆ |
| treewidth min degree ⋆ | average clustering |
| degree mixing matrix ⋆ | resource allocation index |
| average neighbor degree ⋆ | **Connectivity** |
| average degree connectivity ⋆ | subgraph connectivity |
| degree assortativity coef. ⋆ | local node connectivity ⋆ |
| **Centrality** | **Basic properties** |
| katz centrality ⋆ | large clique size ⋆ |
| degree centrality ⋆ | common neighbors |
| closeness centrality ⋆ | number of edges |
| eigenvector centrality ⋆ | number of nodes |
| group degree centrality ⋆ | density ⋆ |
| **Flow property** | local bridges ⋆ |
| min weighted dominating set | |
| min weighted vertex cover | |
| min edge dominating set | |
| min maximal matching | |

Table 1: Graph complexity indices. These indices are manually divided into six categories to ease the presentation and analysis of our results. Indices that are used in our experiments are labeled by the ⋆ symbol. Appendix A provides details on the selection process.

## 2.2 Competence for Gradual Inclusion

Following the core principle of curriculum learning (Bengio et al., 2009), we propose to gradually increase the contribution of harder samples as training progresses. Specifically, we derive the *competence* function $c(t)$ that determines the top fraction of training samples that the model is allowed to use for training at time step $t$. We derive a general form of $c(t)$ by assuming that the rate of competence–the rate by which new samples are added to the current training data–is equally distributed across the remaining training time:

$$\frac{dc(t)}{dt} = \frac{1 - c(t)}{1 - t},\qquad(1)$$

where $t \in [0, 1]$ is the normalized value of the current training time step, with $t = 1$ indicating the time after which the learner is fully competent. Solving this differential equation, we obtain:

$$\int \frac{1}{1 - c(t)} dc(t) = \int \frac{1}{1 - t} c(t),\qquad(2)$$

which results in $c(t) = 1 - \exp(b)(1 - t)$ for some constant $b$. Assuming the initial competence $c(t = 0)$ is $c_0$ and final competence $c(t = 1)$ is 1, we obtain the following linear competence function:

$$c(t) = \min\left(1, 1 - (1 - c_0)(1 - t)\right).\qquad(3)$$

We modify the above function by allowing flexibility in competence so that models can use

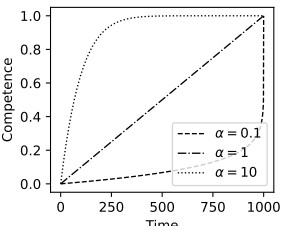

Figure 2: Three competence functions, each imposing a different type of curriculum on GNNs.

larger/smaller fraction of training data than what the linear competence allows at different stages of training. This consideration results in a more general form of competence function:

$$c(t) = \min\left(1, (1 - (1 - c_0)(1 - t))^{\frac{1}{\alpha}}\right),\quad(4)$$

where $\alpha > 0$ specifies the rate of change for competence during training. As Figure 2 shows, a larger $\alpha$ quickly increases competence, allowing the model to use more data after a short initial training with easier samples. We expect such curricula to be more suitable for datasets with lower prevalence of easier samples than harder ones, so that the learner do not spend excessive time on the small set of easy samples at earlier stages of training. On the other hand, a smaller $\alpha$ results in a curriculum that allows more time for learning from easier samples. We expect such curricula to be more suitable for datasets with greater prevalence of easier samples, as it provides sufficient time for the learner to assimilate the information content in easier samples before gradually moving to harder ones.

## 2.3 Spaced Repetition for Ordering Samples

Spaced repetition is a learning technique that involves reviewing and revisiting information at intervals over time. We propose to use spaced repetition to schedule training data for learning. Specifically, we develop schedulers that determine (data from) which complexity indices should be used for training at each time. For this purpose, we learn a *delay* parameter for each index, which signifies the number of epochs by which the usage of data from the index should be delayed before re-introducing the index into the training process. The schedulers dynamically (during training) increase or decrease the delay for indices based on the difficulty of learning their top $c(t)$ samples by the GNN model.

As Algorithm 1 shows, the model first computes complexity indices for training and validation samples, and sorts the samples for each index according to a pre-defined order. All indices are initialized

**Algorithm 1:** TGCL Scheduler

**input :**
    L: Complexity indices
    M: GNN Model
    D: Training data of size $n$
    V: Validation data of size $m$
    S: Index sort order(s)
**output :** Trained model $M^*$

1  $L_i^D \leftarrow$ Complexity of training data based on index $i$
2  $L_i^V \leftarrow$ Complexity of validation data based on index $i$
3  $L_i^D \leftarrow$ sort($L_i^D$, S), $\forall i$
4  $L_i^V \leftarrow$ sort($L_i^V$, S), $\forall i$
5  $\delta_i = 1, \forall i \in$ L #*initialize delay for indices*
6  **for** $t \leftarrow 0$ **to** $E$ **do**
7      current_batch $\leftarrow \{i: \delta_i <= 1\}$
8      delayed_batch $\leftarrow \{i: \delta_i > 1\}$
9      $c(t) \leftarrow$ competence from Eq (4)
10     **for** $i \in$ *current_batch* **do**
11        Train M with top $n \times c(t)$ samples in $L_i^D$
12     **end**
13     **for** $i \in$ *delayed_batch* **do**
14        $\delta_i \leftarrow \delta_i - 1$
15     **end**
16     **for** $i \in$ *current_batch* **do**
17        $s \leftarrow$ top $m \times c(t)$ samples in $L_i^V$
18        $d_i \leftarrow$ loss($s$)
19        $a_i \leftarrow$ prediction_score($s$)
20        $\gamma \leftarrow$ validation_performance($s$)
21        $\delta_i \leftarrow$ compute_delay($i, \eta, \mathbf{d}, \mathbf{a}, \gamma$)
22     **end**
23  **end**

---

**Algorithm 2:** Compute Optimized Delay

**input :**
    $i$: Index
    $\mathbf{d}_i$: Loss vector
    $\mathbf{a}_i$: Probability score vector
    $\eta$: Recall threshold
    $\gamma$: Current model performance on val. data
**output :** $\delta_i$: Delay for index $i$

1  $\hat{\tau}_i \leftarrow$ calculate optimal $\tau$ using (12)
2  $\widehat{\mathbf{t}}_i \leftarrow$ calculate optimal delay using (5)
3  $\delta_i \leftarrow \frac{\sum_j \widehat{\mathbf{t}}_{ij}}{|\widehat{\mathbf{t}}_i|}$ using (6)
4  **return** $\delta_i$

$\widehat{\mathbf{t}}_i$ (Amiri et al., 2017) for the top $c(t)$ samples of each index $i$ in the current batch. We learn the delays such that model performance on the samples is maintained or improved after the delay:

$$\widehat{\mathbf{t}}_i = \arg\max_{\mathbf{t}_{ij}, j \in \mathcal{Q}_i^t} \left( f\left( \frac{\mathbf{d}_{ij} \times \mathbf{t}_{ij}}{\gamma}, \widehat{\tau}_i \right) - \eta \right)^2 \text{(5)}$$

where $\mathcal{Q}_i^t$ is the top $c(t)$ fraction of samples of index $i$ at iteration $t$, $\mathbf{d}_i$ is the instantaneous losses of these samples, $\mathbf{t}_i$ is the delay for these samples (a vector to be learned), $\gamma$ is the performance of the current model on validation data, and $\eta \in (0, 1)$ is the expected model performance on samples in $\mathcal{Q}_i^t$ after the delay. $f()$ is a *non-increasing* function of $\mathbf{x}_i = \frac{\mathbf{d}_i \times \mathbf{t}_i}{\gamma}$, and is responsible for assigning greater delays ($\mathbf{t}_i$) to easier samples (smaller $\mathbf{d}_i$) in stronger networks (greater $\gamma$) (Amiri et al., 2017). Intuitively, (5) estimates the *maximum* delay $\widehat{\mathbf{t}}_i$ for the samples in $\mathcal{Q}_i^t$ such that, with a probability of $\eta$, the performance of the model is maintained or improved for these samples at iteration $e + \widehat{\mathbf{t}}_i$. The hyperparameter $\tau$ controls the rate of decay for $f$, which is optimized using the achieved model performance in hindsight, see §2.3.2. The delay for each index $i$ is obtained by averaging the optimal delays of its top $c(t)$ samples ($\mathcal{Q}_i^t$) as follows:

$$\delta_i = \frac{1}{|\widehat{t}_i|} \sum_{j \in \mathcal{Q}_i^t} \widehat{\mathbf{t}}_{ij}. \tag{6}$$

In addition, indices in the delayed batch are not used for training at current iteration and thus their delays are reduced by one, Line 14, Algorithm 1. We note that, given the improvement gain by the GNN model as training progresses, the above approach is conservative and provides a lower bound of the optimal delays for indices.

### 2.3.2 Scheduling Functions

A good scheduler should assign greater delays to easier samples in stronger models. Therefore, we

with a delay of one, $\delta_i = 1, \forall i$. At every iteration, the model divides the indices into two batches: the *current* batch, those with an estimated delay $\delta_i \leq 1$ iteration; and the *delayed* batch, those with $\delta_i > 1$. Indices in the current batch are those that the scheduler is less confident about their learning by the GNN and those in the delayed batch are indices that are better learned by the GNN. At each iteration, the scheduler prioritizes indices in the current batch by training the GNN using their top $c(t)$ fraction of samples, see (4), while samples of the delayed indices are not used for training. After each iteration, all delay values are updated.

### 2.3.1 Delay Estimation

We develop schedulers that assign greater delays to indices that contain relatively easier samples within their top $c(t)$ samples. Our intuition is that these samples are already learned to a satisfactory degree, thus requires less frequent exposure during training. Delaying such indices can result in better allocation of training resources by preventing unnecessary repetition of already learned samples and potentially directing training toward areas where model's generalization can be further improved.

For this purpose, we first develop a scheduler $f()$ to estimate the optimized *sample-level* delay

can use any non-increasing function of $\mathbf{x}_i = \frac{\mathbf{d}_i \times \mathbf{t}_i}{\gamma}$. We consider the following functions:

$$f_{lap}\left(\mathbf{x}_i, \tau_i\right) = exp(-\mathbf{x}_i \tau_i) \tag{7}$$

$$f_{sec}\left(\mathbf{x}_i, \tau_i\right) = \frac{2}{exp(-\tau_i \mathbf{x}_i^2) + exp(\tau_i \mathbf{x}_i^2)} \tag{8}$$

$$f_{cos}\left(\mathbf{x}_i, \tau_i\right) = \begin{cases} \frac{1}{2}\cos\left(\tau_i \pi \mathbf{x}_i\right) + 1 & \mathbf{x}_i < \frac{1}{\tau_i} \\ 0 & otherwise \end{cases} \tag{9}$$

$$f_{qua}\left(\mathbf{x}_i, \tau_i\right) = \begin{cases} 1 - \tau_i \mathbf{x}_i^2 & \mathbf{x}_i^2 < \frac{1}{\tau_i} \\ 0 & otherwise \end{cases} \tag{10}$$

$$f_{lin}\left(\mathbf{x}_i, \tau_i\right) = \begin{cases} 1 - \tau_i \mathbf{x}_i & \mathbf{x}_i < \frac{1}{\tau_i} \\ 0 & otherwise \end{cases} \tag{11}$$

For each index $i$, we estimate the optimal value of the hyperparameter $\tau_i$ using information from *previous* iteration. Specifically, given the sample loss and validation performance from the previous iteration for the top $c(t - 1)$ samples of index $i$ ($\mathcal{Q}_i^{t-1}$), and the current accuracy of the GNN model on these samples ($\mathbf{p}_i$), we estimate $\tau_i$ as:

$$\widehat{\tau}_i = \arg\min_{\tau_i}\left(f\left(\mathbf{x}_i, \tau_i\right) - \mathbf{p}_i\right)^2, \tag{12}$$
$$\forall_j \in \mathcal{Q}_i^{e-1}, p_{ij} >= \eta.$$

See the steps for delay estimation in Algorithm 2.

## 2.4 Base GNN Models

Our approach can be used to train any GNN model. We consider four models for experiments: Graph-SAGE (Hamilton et al., 2017), graph convolutional network (GCN) (Kipf and Welling, 2017a), graph attention networks (GAT) (Veličković et al., 2018), and graph text neural network (GTNN) (Vakil and Amiri, 2022). GraphSAGE is a commonly-used model that learns node embeddings by aggregating the representation of neighboring nodes through an order-invariant operation. GCN is an efficient and scalable approach based on convolution neural networks which directly operates on graphs. GAT extends GCN by employing self-attention layers to identify informative neighbors while aggregating their information. GTNN extends GraphSAGE for NLP tasks by directly using node representations at the prediction layer as auxiliary information, alleviating information loss in the iterative process of learning node embeddings in GNNs.

## 3 Experiments

### 3.1 Datasets

**Ogbn-arxiv** from Open Graph Benchmark (Hu et al., 2020) is a citation network of computer science articles. Each paper is provided with an embedding vector of size 128, obtained from average word embeddings of the title and abstract of the paper and categorized into one of the 40 categories.

**Cora** (McCallum et al., 2000) is a relatively small citation network, in which papers are categorized into one of the seven subject categories and is provided with a feature word vector obtained from the content of the paper.

**Citeseer** (Kipf and Welling, 2017b) a citation network of scientific articles, in which nodes are classified six classes. We use the same data split as reported in (Zhang et al., 2022).

**Gene, Disease, Phenotype Relation (GDPR)** (Vakil and Amiri, 2022) is a large scale dataset for predicting causal relations between genes and diseases from their text descriptions. Each node in the graph is a gene or disease and an edge represents (causal) relation between genes and diseases.

**Gene Phenotype Relation (PGR)** (Sousa et al., 2019) is a dataset for extracting relations between gene and phenotypes (symptoms) from short sentences. Each node in the graph is a gene, phenotype or disease and an edge represents a relation between its end points. Since the original dataset does not contain a validation split, we generate a validation set from training data through random sampling, while leaving the test data intact. The data splits will be released with our method.

### 3.2 Baselines

In addition to the GNN models described in §2.4, we use the following curriculum learning baselines for evaluation:

**Competence CL (CCL)** (Platanios et al., 2019) is a competence-based CL approach that gradually introduces the data in increasing order of difficulty to the model according to a competence function. The model only works with one difficulty score, which we provide by summing the complexity indices for each training sample.

**SuperLoss (SL)** (Castells et al., 2020) is a CL framework that determines the difficulty of samples by comparing them against the loss value of their corresponding batches. It assigns greater weights to easy samples and gradually introduces harder examples as training progresses.

| | | Node Classification | | | Link Prediction | | |
|---|---|---|---|---|---|---|---|
| **GNN Model** | **Curriculum** | **Ogbn-Arxiv**
**Acc** | **Cora**
**Acc** | **Citeseer**
**Acc** | **GDPR**
**F1** | **PGR**
**F1** | **Average** |
| GTNN | **No-CL** | 71.6±0.1 | 90.4±1.0 | 76.8±0.1 | 84.9±0.3 | 93.9±2.0 | 83.5±0.7 |
| | **CurGraph** | 68.6±0.1 | 86.9±0.8 | 59.9±1.1 | 81.5±1.4 | 73.9±0.2 | 74.2±0.7 |
| | **SL** | 76.1±0.3 | 91.0±0.3 | 77.9±0.8 | 85.0±0.3 | 94.9±0.6 | 85.0±0.4 |
| | **Trend-SL** | 71.7±0.3 | 90.0±0.5 | 77.9±0.1 | 84.9±0.0 | 95.3±0.0 | 84.0±0.2 |
| | **CCL** | 76.4±0.2 | 97.6±0.3 | 76.6±0.7 | 83.6±0.0 | 92.5±0.7 | 85.3±0.4 |
| | **CLNode** | 69.7±0.5 | 75.0±0.1 | 55.7±5.9 | - | - | 66.8±2.2 |
| | **TGCL** | 76.3±0.0 | 96.1±0.8 | 76.7±0.8 | 84.9±0.3 | 93.1±0.7 | **85.4±0.5** |
| GraphSAGE | **No-CL** | 71.4±0.1 | 90.0±0.5 | 75.6±0.5 | 25.4±0.1 | 91.6±1.0 | 70.8±0.4 |
| | **CurGraph** | 69.0±0.2 | 86.7±1.0 | 62.8±1.0 | 65.6±0.5 | 71.3±0.0 | 71.1±0.5 |
| | **SL** | 71.8±0.2 | 89.7±0.5 | 75.5±1.3 | 25.2±0.1 | 91.2±0.6 | 70.7±0.5 |
| | **Trend-SL** | 71.5±0.4 | 88.7±1.3 | 74.6±1.3 | 25.2±0.3 | 91.2±0.6 | 70.3±0.8 |
| | **CCL** | 75.9±0.0 | 96.1±0.3 | 74.4±0.2 | 54.8±0.8 | 88.7±1.6 | 78.0±0.6 |
| | **CLNode** | 60.2±2.4 | 68.9±2.2 | 61.6±4.7 | - | - | 63.5±3.1 |
| | **TGCL** | 75.8±0.3 | 95.8±0.3 | 75.4±0.5 | 56.8±0.1 | 92.4±0.1 | **79.2±0.2** |
| GCN | **No-CL** | 71.8±0.1 | 90.8±1.0 | 76.3±0.6 | 25.2±1.7 | 85.9±1.1 | 70.0±0.9 |
| | **CurGraph** | 70.3±0.3 | 88.2±0.0 | 61.3±1.7 | 70.0±1.4 | 67.2±0.7 | 71.4±0.8 |
| | **SL** | 71.7±0.3 | 89.9±0.8 | 75.5±1.3 | 24.5±1.1 | 84.9±0.4 | 69.3±0.8 |
| | **Trend-SL** | 71.8±0.3 | 90.0±0.5 | 76.3±1.5 | 24.9±0.7 | 84.5±1.0 | 69.5±0.8 |
| | **CCL** | 74.4±0.2 | 91.9±1.0 | 72.5±0.6 | 52.3±0.5 | 84.6±1.5 | 75.1±0.8 |
| | **CLNode** | 60.7±2.0 | 75.5±1.4 | 65.5±0.7 | - | - | 67.2±1.4 |
| | **TGCL** | 74.6±0.2 | 92.3±0.0 | 73.6±0.3 | 53.2±0.4 | 85.2±0.1 | **75.8±0.2** |
| GAT | **No-CL** | 71.0±0.1 | 89.1±0.3 | 76.5±0.7 | 18.8±0.3 | 85.0±0.9 | 68.1±0.5 |
| | **CurGraph** | 69.8±0.2 | 85.6±0.0 | 61.3±1.7 | 92.9±0.2 | 57.5±1.6 | **73.4±0.7** |
| | **SL** | 71.7±0.4 | 88.2±0.0 | 75.4±0.6 | 18.8±0.3 | 84.8±1.1 | 67.8±0.5 |
| | **Trend-SL** | 71.5±0.1 | 89.9±0.3 | 76.4±1.1 | 18.8±0.3 | 85.1±0.5 | 68.3±0.5 |
| | **CCL** | 74.7±0.3 | 92.6±0.0 | 73.3±0.4 | 34.0±0.4 | 84.3±0.9 | 71.8±0.4 |
| | **CLNode** | 64.7±0.3 | 67.8±4.3 | 63.9±0.0 | - | - | 65.5±1.5 |
| | **TGCL** | 74.8±0.0 | 91.3±0.8 | 73.8±0.1 | 34.5±0.6 | 85.9±1.1 | 72.1±0.5 |

Table 2: F1 and Accuracy performance of different curriculum learning models on node classification (Ogbn-Arxiv, Cora, and Citeseer datasets), and link prediction (GDPR and PGR datasets) using GTNN, GraphSAGE, GCN and GAT as base GNN models across three different seeds. All GNN models are initialized with corresponding text embeddings for nodes of each dataset. For the proposed model, TGCL, the best performing kernels for Ogbn-Arxiv, Cora Citeseer, GDPR and PGR are *lap*, *qua*, *sec*, *cos*, and *qua* respectively; we report the top-performing kernel function with average performance and standard deviation over two runs in the Table, bold indicates best performing model, see §3.3 for details.

**CurGraph** (Wang et al., 2021) is a CL approach for GNNs that computes difficulty scores based on the intra- and inter-class distributions of embeddings, realized through a neural density estimator, and develops a smooth-step function to gradually use harder samples in training. We implemented this approach by closely following the paper.

**Trend SL** (Vakil and Amiri, 2022) extends SuperLoss by discriminating easy and hard samples based on their recent loss trajectories. Similar to SL, Trend-SL can be used with any loss function.

**CLNode** (Wei et al., 2023) employs a selective training strategy that estimates sample difficulty based on the diversity in the label of neighboring nodes and identifies mislabeled difficult nodes by analyzing their node features. CLNode implements an easy to hard transition curriculum.

## 3.3 Settings

In the competence function (4), we set the value of $\alpha$ from $[0.2, 5]$. In (5) and (12), we set $\eta$ from $[0.6, 1)$ with step size of 0.1 for link prediction and from $[0.7, 1)$ with the step size of 0.5 for node classification. The best kernel for the datasets in the reported results are *cos*, *qua*, *lap*, *qua*, *sec*, and the best value of $\eta$ is 0.7, 0.9, 0.8, 0.75, 0.9 for GDPR, PGR, Ogbn-Arxiv, Cora and Citeseer respectively. We consider a maximum number of 100 and 500 training iterations for link prediction and node classification respectively. In addition, we order samples for each index in four different ways, ascending/descending (low/high to high/low complexity), and medium ascending/descending (where instances are ordered based on their absolute distance to the standard Normal distribution mean of the complexity scores in ascending/descending or-

der). We evaluate models based on the F1 score for link prediction and accuracy for node prediction task using (Buitinck et al., 2013). Finally, we run all experiments on a single A100 40GB GPU.

### 3.4 Main Results

Table 2 shows the performance of our approach for link prediction and node classification tasks using four GNN models. The performance of all models on link prediction and node classification significantly decreases when we switch from GTNN to any other GNN as encoder, which indicates additional text features as auxiliary information in GTNN are useful for effective learning. See Appendix A.6 for the performance of kernel functions.

For both link prediction and node classification tasks, most curricula improve the performance compared to standard training (No-CL in Table 2). On an average, TGCL performs better than other CL baselines with most GNN models. CurGraph show lower performance than other curricula and No-CL in almost all settings, except when used with GraphSAGE, GCN and GAT on GDPR. The lower overall performance of CurGraph and CLNode may be due to the static order of samples or monotonic curricula imposed by the model, or our implementation of CurGraph. The large performance gains of TGCL on node classification and link prediction datasets against No-CL indicates the importance of the information from difficulty indices, their systematic selection, timely delays, and revisiting indices progressively during training, which help the model generalize better.

## 4 Curricula Introspection

We conduct several analysis on TGCL scheduler to study its behavior and shed light on the reasons for its improved performance. Due to space limitations, we conduct these experiments on one representative dataset from each task, PGR for link prediction and Ogbn-Arxiv for node classification.

### 4.1 Learning Dynamics

For these experiments, we divide training iterations into three phases: Phase-1 (early training, the first 33% of iterations), Phase-2 (mid training, the second 33% of iterations), and Phase-3 (late training, the last 33% of iterations). We report the number of times graph complexity indices appeared in the current batch at each phase. We group indices based on their types and definitions as reported in Table 1 to ease readability.

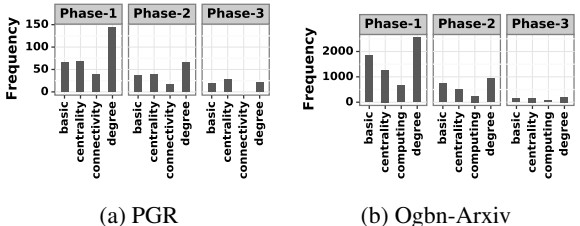

(a) PGR  (b) Ogbn-Arxiv

Figure 3: The number of times each index appeared in the current batch at different phases of training for (a): PGR (link prediction) and (b): Ogbn-Arxiv (node classification) tasks. Phases 1–3 indicate early, mid, and late training respectively quantified by the first, second, and last 33% of training iterations. Degree (local) and centrality (global) based indices are frequently used for link prediction, while degree and basic (local) based indices are frequently used for node classification.

Figures 3a and 3b show the results. The frequency of use for indices follows a decreasing trend. This is expected as in the initial phase the model tends to have lower accuracy in its predictions, resulting in higher loss values. Consequently, the scheduler assigns smaller delays to most indices, ensuring that they appear in the current batch at the early stage of training. However, as the model improves its prediction accuracy, the scheduler becomes more flexible with delays during the latter stages of training. This increased flexibility allows the scheduler to adjust the delay values dynamically and adapt to the learning progress of the model. In addition, the results show that the model focuses on both *local* and *global* indices (degree and centrality respectively) for link prediction, while it prioritizes local indices (degree and basic) over global indices for node classification throughout the training. See Appendix A.4 for detailed results.

### 4.2 TGCL Gains More and Uses Less Data

In standard training, a model uses all its $n$ training examples per epoch, resulting in a total number of $n \times E$ updates. TGCL uses on an average 39.2% less training data for node classification for GTNN model, by strategically delaying indices throughout the training. Figure 4 shows the average number of examples used by different CL models for training across training iteration, computed across all node classification datasets. Our model TGCL uses less data as the training progresses, the standard training (No-CL) and some other curricula such as SL and SL-Trend uses all training data at each iteration. CCL, apart from TGCL, uses less data compared to other CL models. An intriguing observation is that despite both CCL and TGCL are allowed to use

more data as training progresses, TGCL uses less data by strategically delaying indices and avoiding unnecessary repetition of samples that have already been learned, resulting in better training resources and reduced redundancy.

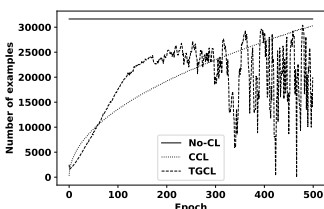

Figure 4: Average number of samples used by different CL frameworks across all node classification datasets. The number remains constant for No-CL, increases for CCL as the training progresses. TGCL uses less data than others by spacing samples over time.

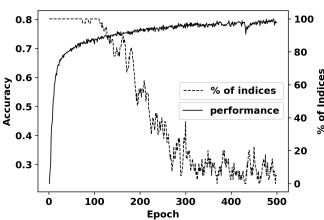

Figure 5: Percentage of indices used for training at every epoch and the average validation accuracy on samples used for training at each iteration on Ogbn-Arxiv. In initial epochs, most indices are frequently used for training until the performance reaches the recall threshold $\eta = .80$, after which the scheduler starts delaying some of the indices. Model prevents repetition of already learned samples, while directing the training towards areas where generalization can be further improved.

In spaced repetition, a spacing effect is observed when the difference between subsequent reviews of learning materials increases as learning progresses. As a result, the percent of the indices used by model for training should decrease as the model gradually becomes stronger. Figure 5 illustrates this behavior exhibited by TGCL. This results demonstrates that the delays assigned by the scheduler effectively space out the data samples over time, leading to an effective training process.

## 4.3 TGCL Prioritizes Linguistics Features

For this analysis, we calculate linguistic indices (detailed in §2.1 and Appendix A.2) from the paper titles in Ogbn-Arxiv. We augment the graph indices with linguistics indices and retrain our top performing model, GTNN, on Ogbn-Arxiv to assess the importance of linguistics indices in the training process. The resulting accuracy is 76.4, which remains unchanged compared to using only graph indices. However, we observe that the model consistently prefers linguistic indices (Coleman Liau Readability and sentence length related indices), followed by the degree based indices, throughout all phases of the training. Figure 6 shows the contribution of linguistic and graph indices in different phases of training. While linguistic indices do not lead to an accuracy beyond 76.4, they are consistently prioritized by TGCL over graph indices. Incorporating additional linguistic indices have the potential to further enhance performance.

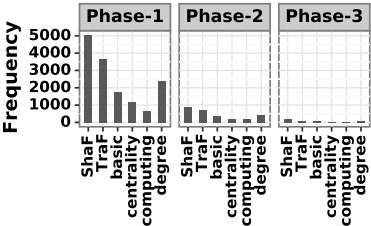

Figure 6: The number of times each index appeared in the current batch at different phases of training for Ogbn-Arxiv when linguistic indices are included. ShaF and TraF are shallow and traditional formulas features described in Appendix A.2.

## 4.4 TGCL Learns Transferable Curricula

We study the transferability of curricula learned by TGCL across datasets and models. For these experiments, we track the curriculum (competence values and indices used for training at every iteration) of a source dataset and apply the curriculum to a target dataset using GTNN as the base model. Table 3 shows learned curricula are largely transferable across dataset, considering the performance of No-CL as the reference. We note that the slight reduction in performance across datasets (compared to the source curricula), can be negligible considering the significant efficiency that can be gained through the adoption of free curricula (39.2% less training data, see §4.2). Table 4 shows the curricula learned by TGCL can be transferred across GNN models, and in some cases improves the performance, e.g., GAT to GCN. Further analysis on these results is the subject of our future works.

## 5 Related Work

In Curriculum learning (CL) (Bengio et al., 2009) data samples are scheduled in a meaningful difficulty order, typically from *easy* to *hard*, for iterative training. In graph machine learning, Wang et al. (2021) introduced CurGraph, a curriculum learning method designed for sub-graph classification. This

| target source | Ogbn-Arxiv | Cora | Citeseer |
|---|---|---|---|
| Ogbn-Arxiv | **76.6** | 94.5 | 75.3 |
| Cora | 76.1 | **96.7** | 76.8 |
| Citeseer | 71.9 | 93.7 | **77.8** |
| No-CL (GTNN) | 71.1 | 91.5 | 75.3 |

Table 3: Performance of curricula transfer across node classification datasets using GTNN. Underline indicates the curricula learned on source dataset.

| target source | GTNN | GraphSAGE | GCN | GAT |
|---|---|---|---|---|
| GTNN | **76.4** | 75.9 | 74.7 | 74.3 |
| GraphSAGE | **76.4** | 75.9 | 74.7 | 74.6 |
| GCN | 75.8 | 75.2 | 74.4 | 74.0 |
| GAT | 76.2 | **76.0** | **75.2** | 74.8 |
| No-CL | 71.1 | 71.5 | 71.8 | 71.8 |

Table 4: Performance of curricula transfer across GNN models on Ogbn-Arxiv. Underline indicates the curricula learned on source GNN model.

model assesses the difficulty of samples by analyzing both intra-class and inter-class distributions of sub-graph embeddings. It then organizes the training instances, by first exposing easier sub-graphs and gradually introducing more challenging ones. Wei et al. (2023) adopted a selective training strategy, targeting nodes with diverse label distributions among their neighbors as particularly challenging to learn. Liu et al. (2023) proposed HSAN, which clusters graphs using curriculum and contrastive learning and measures the difficulty of training pairs using attribute and structural similarity and use weights to select hard negative samples. Wang et al. (2023) proposed an approach called CHEST to improve recommendation using heterogeneous graph data and combine local and global context information to guide curriculum development.

In contrast to static curriculum approaches, Saxena et al. (2019) proposed a dynamic curriculum approach that automatically assigns confidence scores to samples based on their estimated difficulty. However this model requires additional trainable parameters. To address this limitation, Castells et al. (2020) introduced the SuperLoss framework to calculate optimal confidence scores for each instance using a closed-form solution. In (Vakil and Amiri, 2022), we extended SuperLoss to incorporate trend information at the sample level. We utilized loss trajectories to estimate the emerging difficulty of subgraphs and employed weighted sample losses for data scheduling in order to create effective curricula for training GNNs and understanding their learning dynamics.

Current curriculum learning methodologies in NLP rely on data properties, e.g., sentence length, word rarity, or syntactic features (Platanios et al., 2019; Liu et al., 2021), or annotation disagreement (Elgaar and Amiri, 2023); as well as model properties such as training loss and its variations (Graves et al., 2017; Amiri et al., 2017) to sequence data samples for training. Elgaar and Amiri (2023) developed a curriculum discovery framework based on prior knowledge of sample difficulty, utilized annotation entropy and loss values. They concluded that curricula based on easy-to-hard or hard-to-easy transition are often at the risk of under-performing, effective curricula are often non-monotonic, and curricula learned from smaller datasets perform well on larger datasets.

Other instances of curriculum learning for textual data have primarily centered on machine translation and language comprehension. For instance, Agrawal and Carpuat (2022) introduced a framework for training non-autoregressive sequence-to-sequence models for text editing. Additionally, Maharana and Bansal (2022) designed various curriculum learning approaches where the teacher model assesses the difficulty of each training example by considering factors such as question-answering probability, variability, and out-of-distribution measures. Other notable work in various domain includes (Graves et al., 2017; Jiang et al., 2018; Castells et al., 2020; Settles and Meeder, 2016; Amiri et al., 2017; Zhang et al., 2019; Lalor and Yu, 2020; Xu et al., 2020; Kreutzer et al., 2021) which have contributed to its broader adoption.

## 6 Conclusion and Future Work

We introduce a novel curriculum learning approach for text graph data and graph neural networks, inspired by spaced repetition. By leveraging text and graph complexity formalisms, our approach determines the optimized timing and order of training samples. The model establishes curricula that are both data-driven and model- or learner-dependent. Experimental results demonstrate significant performance improvements in node classification and link prediction tasks when compared to strong baseline methods. Furthermore, our approach offers potential for further enhancements by incorporating additional complexity indices, exploring different scheduling functions and model transferability, and extending its applicability to other domains.

## Broader Impacts

The advancements in curriculum learning signal a promising direction for the optimization of training processes within NLP and graph data. Based on the principles of "spaced repetition" and text and graph complexity measures, the proposed work enhances the efficiency of training and improves model generalization capabilities. This is particularly crucial for applications reliant on graph representations of text, such as social network analysis, recommendation systems, and semantic web. Furthermore, the method's ability to derive transferable curricula across different models and datasets suggests a more applicable strategy, potentially enabling seamless integration and deployment across varied NLP applications and domains.

## Limitation

The proposed approach relies on the availability of appropriate complexity formalisms. If the selected indices do not capture the desired complexity, the curricula may not be optimally designed. The approach primarily focuses on text graph data and graph neural networks, and the results may not directly apply to other types of data or architectures. The estimation of optimized time and order for training samples introduces additional computational overhead. This can be a limitation in scenarios where real-time training is required, e.g., in processing streaming data of microposts.

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

## A Appendix

### A.1 Graph Indices Definition

Below are the list of 26 indices which we consider for TGCL. All these indices are computed on the subgraph of the node or an edge. These definition and code to calculate the indices, we used Networkx package (Hagberg et al., 2008).

- **Degree:** The number of immediate neighbors of a node in a graph.

- **Treewidth min degree:** The treewidth of an graph is an integer number which quantifies, how far the given graph is from being a tree.

- **Average neighbor degree:** Average degree of the neighbors of a node is computed as:

$$\frac{1}{|\mathcal{N}_i|} \sum_{j \in \mathcal{N}_i} k_j$$

  where $\mathcal{N}_i$ is the set of neighbors of node $i$ and $k_j$ is the degree of node $j$.

- **Degree mixing matrix:** Given the graph, it calculates joint probability, of occurrence of node degree pairs. Taking the mean, gives the degree mixing value representing the given graph.

- **Average degree connectivity:** Given the graph, it calculates the average of the nearest neighbor degree of nodes with degree $k$. We choose the highest value of $k$ obtained from the calculation and used its connectivity value as the complexity index score.

- **Degree assortativity coefficient:** Given the graph, assortativity measures the similarity of connections in the graph with respect to the node degree.

- **Katz centrality:** The centrality of a node, $i$, computed based on the centrality of its neighbors $j$. Katz centrality computes the relative influence of a node within a network by measuring taking into account the number of immediate neighbors and number of walks between node pairs. It is computed as follows:

$$x_i = \alpha \sum_j A_{ij} x_j + \beta$$

where $x_i$ is the Katz centrality of node $i$, $A$ is the adjacency matrix of Graph $G$ with eigenvalues $\lambda$. The parameter $\beta$ controls the initial centrality and $\alpha < 1 / \lambda_{max}$.

- **Degree centrality:** Given the graph, the degree centrality for a node is the fraction of nodes connected to it.

- **Closeness centrality:** The closeness of a node is the distance to all other nodes in the graph or in the case that the graph is not connected to all other nodes in the connected component containing that node. Given the subgraph and the nodes, added the values of the nodes to find the complexity index value.

- **Eigenvector centrality:** Eigenvector centrality computes the centrality for a node based on the centrality of its neighbors. The eigenvector centrality for node i is $Ax = \lambda x$. where $A$ is the adjacency matrix of the graph $G$ with eigenvalue $\lambda$.

- **Group Degree centrality:** Group degree centrality of a group of nodes S is the fraction of non-group members connected to group members.

- **Ramsey R2:** This computes the largest clique and largest independent set in the graph $G$. We calculate the index value by multiplying number of largest cliques to number of largest independent set.

- **Average clustering:** The local clustering of each node in the graph $G$ is the fraction of triangles that exist over all possible triangles in its neighborhood. The average clustering coefficient of a graph $G$ is the mean of local clusterings.

- **Resource allocation index:** For nodes $i$ and $j$ in a subgraph, the resource allocation index is defined as follows:

$$\sum_{k \in (\mathcal{N}_i \bigcap \mathcal{N}_j)} \frac{1}{|\mathcal{N}_k|},$$

  which quantifies the closeness of target nodes based on their shared neighbors.

- **Subgraph density:** The density of an undirected subgraph is computed as follows:

$$\frac{e}{v(v-1)},$$

where $e$ is the number of edges and $v$ is the number of nodes in the subgraph.

- **Local bridge:** A local bridge is an edge that is not part of a triangle in the subgraph. We take the number of local bridges in a subgraph as a complexity score.

- **Number of nodes:** Given the graph $G$, number of nodes in the graph is chosen as the complexity score.

- **Number of Edges:** Given the graph $G$, number of edges in the graph is chosen as the complexity score.

- **Large clique size:** Given the graph $G$, the size of a large clique in the graph is chosen as the complexity score.

- **Common neighbors:** Given the graph and the nodes, it finds the number of common neighbors between the pair of nodes. We chose number of common neighbors as the complexity score.

- **Subgraph connectivity:** is measured by the *minimum* number of nodes that must be removed to disconnect the subgraph.

- **Local node connectivity:** Local node connectivity for two non adjacent nodes s and t is the minimum number of nodes that must be removed (along with their incident edges) to disconnect them. Given the subgraph and the nodes, gives the single value which we used as complexity score.

- **Minimum Weighted Dominating Set**: For a graph $G = (V, E)$, the weighted dominating set problem is to find a vertex set $\mathcal{S} \subseteq V$ such that when each vertex is associated with a positive number, the goal is to find a dominating set with the minimum weight.

- **Weighted vertex cover index:** The weighted vertex cover problem is to find a vertex cover $\mathcal{S}$–a set of vertices that include at least one endpoint of every edge of the subgraph–that has the minimum weight. This index and the weight of the cover $\mathcal{S}$ is defined by $\sum_{s \in \mathcal{S}} w(s)$, where $w(s)$ indicates the weight of $s$. Since $w(s) = 1, \forall s$ in our unweighted subgraphs, the problem will reduce to finding a vertex cover with minimum cardinality.

- **Minimum edge dominating set:** Minimum edge dominating set approximate solution to the edge dominating set.

- **Minimum maximal matching:** Given a graph G = (V,E), a matching M in G is a set of pairwise non-adjacent edges; that is, no two edges share a common vertex. That is, out of all maximal matchings of the graph G, the smallest is returned. We took the length of the set as the complexity index.

## A.2 Linguistics Indices

Below are the list of linguistic (Lee et al., 2021b) indices used in our experiments. We follow (Lee et al., 2021b) to measure all scores.

**Traditional Formulas (TraF)** These features computes the readability score of the given text based on the content, complexity of its vocabulary and syntactic information. Readability can be defined as the ease with which a reader can understand a written text.

- **Gunning Fog Count score:** The Gunning fog index is a readability test for English writing. It commonly used to confirm that text can be read easily by the intended audience. It is computed as follows:
$0.4 \left[ \left( \frac{words}{sentences} \right) + 100 \left( \frac{complexwords}{words} \right) \right]$
where *words* is the number of words, *sentences* is the number of sentences, and *complexwords* is the number of complex words

- **New Automated readability index:** The automated readability index is a readability test for texts, which determines the understandability of a text. It is computed as follows:

$4.71 \left[ \left( \frac{characters}{words} \right) + 0.5 \left( \frac{words}{sentences} \right) \right]$

where *characters* is the number of letters and numbers, *words* is the number of spaces, and *sentences* is the number of sentences.

- **Flesch Kincaid Grade level:** This readability test is design to determine how difficult is the given text to understand. It can be computed as follows:

$0.39 \left[ \left( \frac{words}{sentences} \right) + 11.8 \left( \frac{syllables}{words} \right) \right]$

where *words* is the total number of words, *sentences* is the total number of sentences, and *syllables* is the total number of syllables

- **Linsear Write Formula score:** It is a readability metric for text originally developed to calculate the readability of technical manuals. It can be computed as follows:

---

**Algorithm 3:** Compute Linsear Write score

1   Initialize $r = 0$
2   For each *easy word*, defined as word with 2 syllables or less $r = r + 1$
3   For each *hard word*, defined as word with 3 syllables or more $r = r + 3$

$$r = \frac{r}{sentences}$$

    where *sentences* = number of sentences in 100 word sample
4   if $r > 20$, $LinsearWritescore = \frac{r}{2}$
5   if $r =< 20$, $LinsearWritescore = \frac{r}{2} - 1$

---

- **Coleman Liau Readability Score:** The Coleman–Liau index is calculated as follows:

$$0.0588 * L - 0.296 * S - 15.8$$

where L is the average number of letters per 100 words, S is the average number of sentences per 100 words

- **SMOG Index:** SMOG index can be calculated as follows:

$$1.0430 * (polysyllables * \frac{30}{sentences})^{1/2} + 3.1291$$

**Shallow Features (ShaF)** These features captures the surface level difficulties. Features used are as follows:

- **Average count of characters per token:** The average count of characters per token is taken as the complexity score.

- **Average count of characters per sentence:** The average count of characters per sentence is taken as the complexity score.

- **Average count of syllables per token:** The average count of syllables per token is taken as the complexity score.

- **Average count of syllables per sentence:** The average count of characters per syllables is taken as the complexity score.

- **Sentence length:** computed by count of token per sentence

- **Token sentence ratio:** computed by the log of total count of tokens divided by the log of total count of sentences.

- **Token sentence multiply:** computed by the total count of tokens multiply by the total count of sentences, and its square root.

## A.3   TGCL Exploits All Ranking Orders

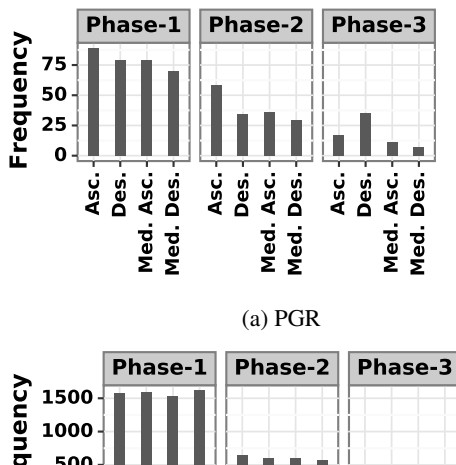

(a) PGR

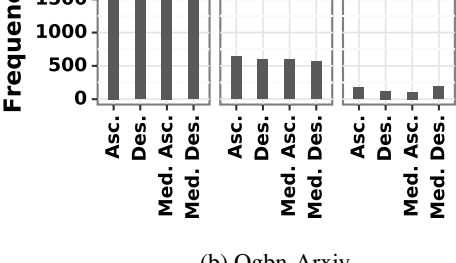

(b) Ogbn-Arxiv

Figure 7: The distribution of indices used for training based on their ordering strategy. Overall, scheduler (a): relies more on ascending order in the case of PGR and (b): relies almost equally on all order types in the case of Ogbn-Arxiv. The use of medium ascending order is in line with recent studies showing the importance of using medium-level samples for effective training (Swayamdipta et al., 2020).

Figures 7a and 7b show the distribution of indices used for training based on to their ordering strategy (see §2.1 and Algorithm 1) during different phases of training. As mentioned before, complexity scores can be sorted in four ways: ascending, descending, medium ascending, and medium descending orders, which represent easy-to-hard or hard-to-easy order types. The results on the PGR dataset show that in the initial phase of training the scheduler uses all order types, while emphasizing most on ascending followed by descending orders. And, in the mid and late training phases, the model prioritizes the ascending difficulty order over the other orders with a fairly larger difference in usage. The results on Ogbn-Arxiv show that TGCL relies equally on all order types during its training with a slightly greater emphasis on descending order at the latter stages of training. The relatively significant use of medium ascending order, especially

at the early stage of training, is in line with recent studies showing the importance of using medium-level samples for effective training (Swayamdipta et al., 2020).

## A.4 Fine-grained Index Analysis

Figure 8 shows fine-grained analysis for Ogbn-Arxiv when linguistic indices are included along with the graph indices. In the Phase-1, scheduler focuses on all the indices with more frequency of SraF based indices from linguistics, and Ramsey and degree based indices from graph features. In the Phase-2, the overall use of all the indices is reduced and it focuses more on readability indices (TraF) from linguistics features, and uses Ramsey R2 more from the graph indices. In Phase-3, scheduler uses very less indices at the end of the training and focuses on the average count of characters per sentence from linguistic features and degree assortaivity coefficient from the graph indices.

As shown in the Figure 9 for PGR dataset, centrality and degree based indices are used more frequently. Closeness centrality, density, and degree assortativity coefficient indices are used more frequently in Phase-1 and Phase-2, initial part and middle part of training. In the final phase of the training Phase-3, scheduler focuses on closeness centrality and degree connectivity based indices more frequently.

As shown in the Figure 10, for Ogbn-Arxiv dataset, basic and degree based indices are used more frequently. In the initial phase of training, as the threshold is high ($\eta$) scheduler uses all available indices. In the Phase-2, scheduler uses degree centrality, Ramsey R2 and degree assortativity coefficient more frequently. In Phase-3 scheduler uses local bridge, degree and degree centrality more frequently compare to the other indices.

## A.5 Selection Process for Complexity Indices

To avoid the over representation of similar indices, we group indices based on their similarity. For this purpose, we compute the Pearson Co-relation coefficient between complexity scores of each pair of indices and create an $n \times n$ correlation matrix (where, $n$ is the total number of indices). We use this co-relation matrix as an input to K-means and empirically group the indices into 10 clusters. We randomly select one metric from each cluster for use in our curriculum framework. See indices with $\star$ labels in Table 1. Here, $\star$ indicates the indices

were used in one of the dataset used in the experiments.

## A.6 Detailed results

Table 5 shows the performance of TGCL model with different kernel functions for all the datasets on GTNN as the base model.

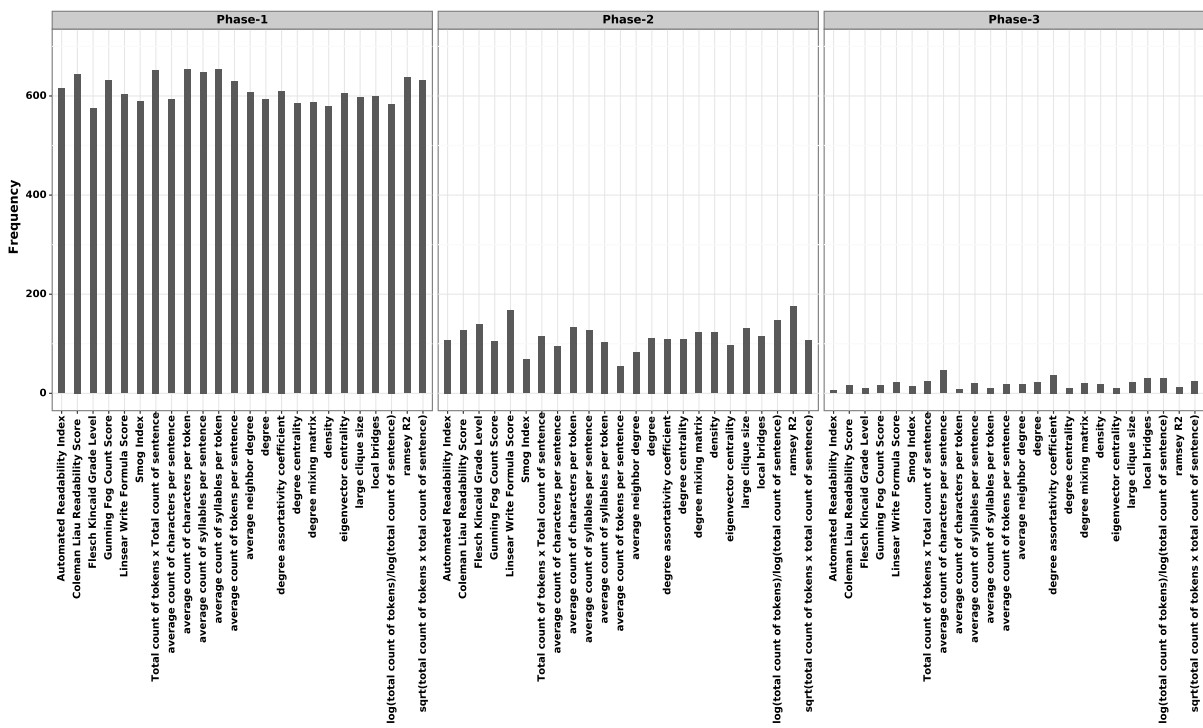

Figure 8: Fine grained indices priority for Ogbn-Arxiv with linguistic indices

| GNN Model | TGCL Kernel | Ogbn-Arxiv Acc | Cora Acc | Citeseer Acc | GDPR F1 | PGR F1 |
|---|---|---|---|---|---|---|
| GTNN | cos | 76.3 | 95.9 | 76.1 | 85.4 | 94.5 |
| | gau | 76.2 | 95.6 | 75.8 | 85.1 | 94.5 |
| | lap | 76.4 | 95.2 | 76.1 | 85.3 | 89.2 |
| | lin | 75.7 | 95.6 | 76.1 | 84.4 | 87.9 |
| | sec | 76.2 | 95.6 | 77.8 | 84.3 | 94.5 |
| | qua | 75.9 | 96.7 | 77.7 | 85.0 | 93.2 |

Table 5: F1 and Accuracy performance of **TGCL** model for different kernel functions on node classification (Ogbn-Arxiv, Cora, and Citeseer datasets), and link prediction (GDPR and PGR datasets) using GTNN as base GNN model.

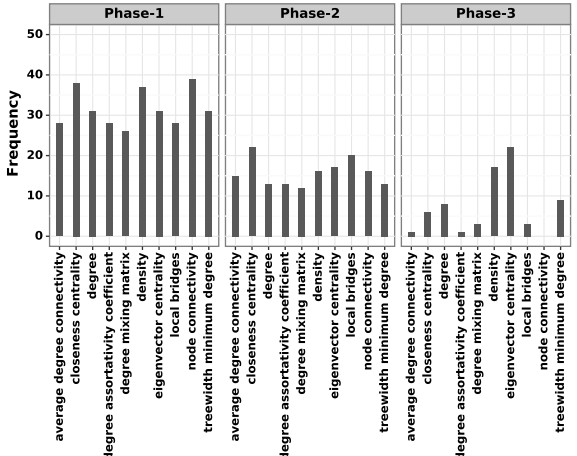

Figure 9: Fine grained indices priority for edge prediction task.

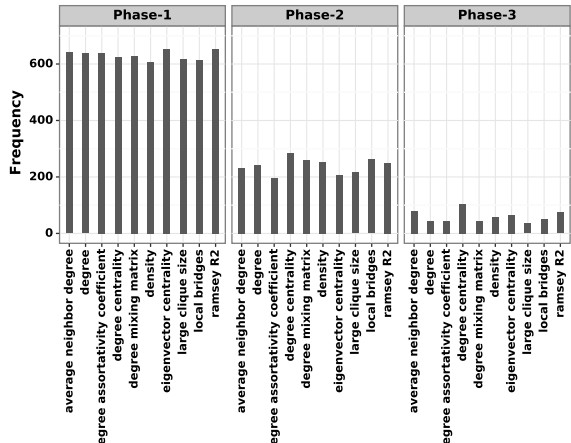

Figure 10: Fine grained indices priority for node prediction task.