# OpenReview forum: "Complexity-Guided Curriculum Learning for Text Graphs"
_EMNLP/2023/Conference — EMNLP 2023 Findings_

### Official Review · Reviewer_G4id · 2023-08-02

**Typos Grammar Style And Presentation Improvements:** 1.	Grammar
**Soundness:** 3

**Excitement:**

3: Ambivalent: It has merits (e.g., it reports state-of-the-art results, the idea is nice), but there are key weaknesses (e.g., it describes incremental work), and it can significantly benefit from another round of revision. However, I won't object to accepting it if my co-reviewers champion it.

**Paper Topic And Main Contributions:**

This paper proposed a curriculum learning approach for text graphs. The main hypothesis of the paper is that the graph and text complexity have impact on the learning process. Following the existing curriculum learning paradigm, the authors employed various graph and text complexity indices to quantify the difficulty of a training instance and used a competence function to include examples from easy to hard. In my understanding, the main contributions of the paper are two-fold. First, a training schedular is proposed to select the graph/text complexity indices that may improve the learning process in each epoch. The training schedular is general and is applicable to general curriculum learning problems. Second, the paper finds that both text and graph complexity indices are useful in curriculum learning for text graphs and text complexity indices are more frequently used.

**Questions For The Authors:**

1.	Why the existing methods can’t be used for text graphs? What are the research gaps?
2.	What is the intuition of Equation 5?
3.	How many independent runs have been conducted to get the accuracy and F1 in Table 2?
4.	How the proposed method enables transferability of the learned curricula.

**Reasons To Accept:**

1.	The proposed training scheduler can be generalized to other curriculum learning scenarios. The novelty of the work lies in the utilization of various complexity indices and the algorithms for selecting/delaying the use of some indices during the training process.
2.	The paper analyses the importance of different indices during different periods of the training process.

**Reasons To Reject:**

1.	The paper lacks a clear motivation for considering text graphs. Except for the choice of complexity indices, which can easily be changed according to the domain, the proposed method is general and can be applied to other graphs or even other types of data. Moreover, formal formulations of text graphs and the research question are missing in the paper.
2.	Several curriculum learning methods have been discussed in Section 1. However, the need for designing a new curriculum learning method for text graphs is not justified. The research gap, e.g., why existing methods can’t be applied, is not discussed.
3.	Equations 7-11 provide several choices of function f. However, there is no theoretical analysis or empirical experiments to advise on the choice of function f.
4.	In the overall performance comparison (Table 2), other curriculum learning methods do not improve performance compared to No-CL. These results are not consistent with the results reported in the papers of the competitors. At least some discussions about the reason should be included. In addition, it is unclear how many independent runs have been conducted to get the accuracy and F1. What are the standard deviations?
5.	Although experimental results in Table 3 and Table 4 show that the performance remains unchanged, it is unclear how the transfer of knowledge is done in the proposed method. An in-depth discussion of this property should strengthen the soundness of the paper.
6.	In line 118, the authors said the learned curricula are model-dependent, but they also said the curricula are transferrable across models. These two statements seem to be contradictory.

**Reproducibility:**

3: Could reproduce the results with some difficulty. The settings of parameters are underspecified or subjectively determined; the training/evaluation data are not widely available.

**Reviewer Confidence:**

3: Pretty sure, but there's a chance I missed something. Although I have a good feel for this area in general, I did not carefully check the paper's details, e.g., the math, experimental design, or novelty.

---

> ### Author Rebuttal · Authors · 2023-08-25
>
> We thank the reviewer for their time and comments.
>
> ***Q :*** Why the existing methods can’t be used for text graphs? What are the research gaps? Moreover, formal formulations of text graphs and the research question are missing in the paper. The need for designing a new curriculum learning method for text graphs is not justified.
>
> ***A :*** Existing methods are applicable to text graphs, please see the range of baseline models in our paper. The existing gap in current research, however, is that existing models are often limited to a single criterion for assessing sample difficulty, typically instantaneous training loss and its variations. This approach overlooks the nature of difficulty: difficulty is a condition that can be realized from multiple perspectives, can vary across a continuum for different models, and can dynamically change as the model improves. To bridge this gap, we propose to utilize graph and text complexity formalisms, which provide a more natural and holistic framework for understanding sample complexity. To our knowledge, no existing model utilizes these formalisms for curriculum learning on text-graphs. Our model addresses these gaps in existing research. In addition, we note that text graphs in existing literature are defined as structured representations that interconnect textual elements.
>
> ***Q :*** The paper lacks a clear motivation for considering text graphs. Except for the choice of complexity indices, which can easily be changed according to the domain, the proposed method is general and can be applied to other graphs or even other types of data.
>
> ***A :*** As mentioned by the reviewer our approach is applicable to other graphs or genres of data, which we consider as a key strength indicating the potential of the model beyond NLP. However, our research is rooted in NLP, which motivates our use of text and graph complexity indices and text-graph data.
>
> ***Q :*** What is the intuition of Equation 5?
>
> ***A :*** Equation 5 gradually deemphasizes redundant and already-learned complexity indices by increasing intervals between their subsequent use in training. It dynamically and iteratively learns the duration of these intervals for indices (see the delay parameter, $t_i$) according to the difficulty of their sample ($d_i$) and learning dynamics of the model ($\gamma$).
>
> ***Q :*** How many independent runs have been conducted to get the accuracy and F1 in Table 2?
>
> ***A :*** In our experiments, we fixed the random seed and ran all the models once to report the accuracy and F1 in Table 2. However, to answer the reviewer's question, we ran our model with GTNN as a base model on Arxiv dataset with 3 different seeds and the average performance is 76.04 ± 0.06 (mean, SD). We will report results with 3 seeds for all models and datasets in Appendix.
>
> ***Q :*** In the overall performance comparison (Table 2), other curriculum learning methods do not improve performance compared to No-CL.
>
> ***A :*** Our baseline curriculum learning approaches including CurGraph, CLNode, and CCL occasionally perform worse than no curriculum learning setting (No-CL). This could be attributed to the monotonic nature (i.e., easy-to-hard transition) of these curricula. Recent research shows that monotonic curricula “are often at the risk of underperforming” (Elgaar, et al., ACL’23). On the other hand, non-monotonic methods such as SL and Trend-SL baselines and our model (TGCL), often outperform No-CL across all based GNN models and datasets, showing their effectiveness.
>
> ***Q :*** How the proposed method enables transferability of the learned curricula? In line 118, the authors said the learned curricula are model-dependent, but they also said the curricula are transferrable across models. These two statements seem to be contradictory.
>
> ***A :*** The primary objective of our work is to develop curricula for text-graph data by leveraging graph and linguistic complexity indices. Transferability experiments show that these learned curricula can be applied to other datasets or GNN models, resulting in significant efficiency gain and comparable performance. Although investigating the reasons falls beyond the primary scope of our current paper, we conjecture that the transferability of the learned curricula derived from complexity formalisms can be attributed to two key factors: (a): shared topological properties among graphs within the same or different domains, such as prevalent features and characteristics in citation networks or power law distribution of node degrees in most graphs, and (b): the fundamental message-passing architecture inherent to most GNN models, which could associates their learning dynamics. However, further analysis on these ideas is the subject of the future work as mentioned in Line# 522.
>
> ***Q :*** Equations 7-11 provide several choices of function f. However, there is no theoretical analysis or empirical experiments to advise on the choice of function f.
>
> ***A :*** As mentioned in Line 266, our model is applicable to any non-increasing function f (sample scheduler), and Equations 7—11 exemplify such functions. We reported the best performing function for each dataset, considering the space limit. However, we note that model performance varies across these functions as they differ in their flexibility in delaying samples for training. Overall, quadratic, laplacian and secant functions are more effective (and more flexible in delaying indices) while linear and cosine functions are less effective (and also less flexible in delaying indices). We will report detailed results of these functions in Appendix.

---

### Official Review · Reviewer_xYVm · 2023-08-02

**Soundness:** 3

**Excitement:**

4: Strong: This paper deepens the understanding of some phenomenon or lowers the barriers to an existing research direction.

**Paper Topic And Main Contributions:**

In this submission, they propose a data scheduler for text graph tasks.

First, they evaluate the difficulty of data samples based on 26 graph complexity indices (e.g., Degree, Treewidth min degree, Average neighbor degree ) and 14 traditional and shallow linguistics indices (e.g. Readability, Average count of characters per token, Average count of characters per sentence).

Second, they design a  competence function to introduce training samples to GNNs gradually. Specifically, they use Spaced Repetition to determine which complexity indices should be used for training at each time.





**Reasons To Accept:**

1. They take full account of serval difficulty features and design a data scheduler, see the above part.

2. They conduct sufficient experiments on node classification datasets and link prediction datasets.

3. They also analyze the effect of different difficulty features.


**Reasons To Reject:**


1. There are many parameters in the formulas in Section 2, but there are some that are not explained in detail.

2.  Section 2.3 and Algorithm 1 are difficult to follow.

**Reproducibility:**

3: Could reproduce the results with some difficulty. The settings of parameters are underspecified or subjectively determined; the training/evaluation data are not widely available.

**Reviewer Confidence:**

3: Pretty sure, but there's a chance I missed something. Although I have a good feel for this area in general, I did not carefully check the paper's details, e.g., the math, experimental design, or novelty.

---

> ### Author Rebuttal · Authors · 2023-08-25
>
> We thank the reviewer for their time and comments.
>
> ***Q :*** Some parameters in formulas in Section 2 are not explained in detail.
>
> ***A :*** We thank the reviewer for the comment. However, we cross checked all the equations and found that all parameters are explained in the paper. In order to assist the reviewer, we reiterate these parameters and provide concise definitions below:
>
> Section 2.2 Competence for Gradual Inclusion
>
> * c(t) indicates the fraction of data that can be used for training (i.e., competence) at time t
> * $c_0$ is the initial competence value
> * $\alpha$ > 0 controls the shape of the competence function
>
> Section 2.3 Spaced Repetition for Ordering Samples
>
> * $t_i$ is the current delay of the complexity index i
> * $\hat{t}_i$ is the predicted/optimized delay for complexity index i
> * $\gamma$ is the performance of the current model on the validation data
> * $d_i$ is the vector of instantaneous losses of the top examples ranked by complexity index i
> * $\eta$ is the expected performance of the model during training, which is used to calculate optimal delay
> * $\hat{\tau}_i$ is the optimal value for the hyperparameter $\tau$ used in the kernel functions.
>
> ***Q :*** Section 2.3 and Algorithm 1 are difficult to follow.
>
> ***A :*** The key idea of Section 2.3 and algorithm 1: given that different complexity indices present different orders of training samples, the idea is to strategically deemphasize redundant and already-learned (i.e., easy) complexity indices by increasing intervals between their subsequent use in training. The model dynamically and iteratively learns the duration of these intervals for indices (see the delay parameter) according to the difficulty of their top k sample and learning dynamics of the model.

---

### Official Review · Reviewer_8ekE · 2023-08-04

**Typos Grammar Style And Presentation Improvements:** None.
**Soundness:** 3

**Excitement:**

3: Ambivalent: It has merits (e.g., it reports state-of-the-art results, the idea is nice), but there are key weaknesses (e.g., it describes incremental work), and it can significantly benefit from another round of revision. However, I won't object to accepting it if my co-reviewers champion it.

**Missing References:**

None.

**Paper Topic And Main Contributions:**

This paper presents a curriculum learning approach for effective training with text graph data. The authors hypothesize that existing knowledge about text and graph complexity can inform better curriculum development for text graph data. Motivated by this rationale, a complexity-guided CL approach is proposed for text graphs (TGCL). Specifically, TGCL employs multiview complexity formalisms to space training samples over time for iterative training. The experimental results demonstrate the effectiveness of the proposed method.

**Questions For The Authors:**

None.

**Reasons To Accept:**

1. The writing is clear and easy to follow.
2. This paper proposes a new curriculum learning framework for training GNNs on text graph data and provides insights into what complexity formalisms are learned by GNNs during their training.

**Reasons To Reject:**

I am not an expert in the field of curriculum learning, and I have the following concerns:
1. It seems that the proposed method needs to select appropriate complexity formalisms based on human experience.
2. Recent research on learning from text graphs has achieved impressive performance by jointly [1] or iteratively [2] training GNN and LM. Can the proposed complexity-guided CL approach be well applied to such joint or iterative framework?

[1] Chien E, Chang W C, Hsieh C J, et al. Node feature extraction by self-supervised multi-scale neighborhood prediction. ICLR 2022.

[2] Zhao J, Qu M, Li C, et al. Learning on large-scale text-attributed graphs via variational inference. ICLR 2023.

**Reproducibility:**

3: Could reproduce the results with some difficulty. The settings of parameters are underspecified or subjectively determined; the training/evaluation data are not widely available.

**Reviewer Confidence:**

2: Willing to defend my evaluation, but it is fairly likely that I missed some details, didn't understand some central points, or can't be sure about the novelty of the work.

---

> ### Author Rebuttal · Authors · 2023-08-25
>
> We thank the reviewer for their time and comments.
>
> ***Q :*** It seems that the method needs to select appropriate complexity formalisms based on human experience.
>
> ***A :*** The process of establishing complexity formalisms for graph and text is an active area of research. In the current literature, complexity measures are derived through both manual development (by experts) and data-driven automated extraction, which reflects the ongoing exploration in this area. Our model can work with any number of current and future complexity indices and can dynamically emphasize/deemphasize them for effective training. In addition, we note that our model utilizes both the complexity of samples and the learning dynamics of a downstream model to guide training.
>
> ***Q :*** Can the proposed complexity-guided CL approach be well applied to recent techniques that jointly [1] or iteratively [2] train GNN and LM models?
>
> ***A :*** The proposed curricula (and the ones developed by our baselines) strategically arrange examples for iterative training and, as training paradigms, can be applied to a wide range of neural models including the joint or iterative frameworks suggested by the reviewer. Therefore the key evaluation aspect in our paper centers on comparative assessment among different curricula. While we incorporated four distinct base GNN architectures in our experiments, it is important to acknowledge that including other architectures would greatly benefit from in-depth exploratory research, which falls beyond the primary scope of our current paper. We will highlight this point in the paper.

---

### Meta-Review · Area_Chair_7hZW · 2023-09-19

**Recommendation:** 4

**Metareview:**

This paper proposes a curriculum learning (CL) approach (especially a data scheduler) for effective training with text graph data.

Pros:
1) The proposed curriculum learning method is overall novel and can generalize to other scenarios besides text graphs.
2) The experiment on several node classification and link prediction datasets is sufficient.

Cons:
1) In the main results (Table 2), some recent baselines cannot outperform the No-CL method, which is also mentioned by one reviewer. The authors' rebuttal cannot fully convince me because the most recent baseline CLNode performs worse than No-CL in almost all the datasets. I wonder whether the experimental setting is fair for the proposed method and all the basellines.
2) The connection between the proposed CL method and text graph tasks should be described more clearly. Since this paper targets at a specific data type, the design of the proposed method should include some features that can deal with the core problems in text graph data.

---

### Decision · Program_Chairs · 2023-10-07

**Decision:**

Accept-Findings

**Comment:**

This paper proposes a curriculum learning (CL) approach (especially a data scheduler) for effective training with text graph data.

Pros:
1) The proposed curriculum learning method is overall novel and can generalize to other scenarios besides text graphs.
2) The experiment on several node classification and link prediction datasets is sufficient.

Cons:
1) In the main results (Table 2), some recent baselines cannot outperform the No-CL method, which is also mentioned by one reviewer. The authors' rebuttal cannot fully convince me because the most recent baseline CLNode performs worse than No-CL in almost all the datasets. I wonder whether the experimental setting is fair for the proposed method and all the basellines.
2) The connection between the proposed CL method and text graph tasks should be described more clearly. Since this paper targets at a specific data type, the design of the proposed method should include some features that can deal with the core problems in text graph data.